# Recombinant Strains of Human Parechovirus in Rural Areas in the North of Brazil

**DOI:** 10.3390/v11060488

**Published:** 2019-05-29

**Authors:** Élcio Leal, Adriana Luchs, Flávio Augusto de Pádua Milagres, Shirley Vasconcelos Komninakis, Danielle Elise Gill, Márcia Cristina Alves Brito Sayão Lobato, Rafael Brustulin, Rogério Togisaki das Chagas, Maria de Fátima Neves dos Santos Abrão, Cássia Vitória de Deus Alves Soares, Fabiola Villanova, Steven S. Witkin, Xutao Deng, Ester Cerdeira Sabino, Eric Delwart, Antônio Charlys da Costa

**Affiliations:** 1Instituto de Ciências Biológicais, Universidade Federal do Pará, Pará 66075-000, Brazil; fevface@gmail.com; 2Laboratório de Doenças Entéricas, Centro de Virologia, Instituto Adolfo Lutz, São Paulo 01246-000, Brazil; driluchs@gmail.com; 3LIM/46, Faculdade de Medicina, Universidade de São Paulo, São Paulo 01246-903, Brazil; flaviomilagres@uft.edu.br; 4Secretaria de Saúde do Tocantins, Tocantins 77453-000, Brazil; eumarciaalvesbrito@gmail.com (M.C.A.B.S.L.); eu3rafael@gmail.com (R.B.); chagastogisaki@hotmail.com (R.T.d.C.); fatima_abrao@yahoo.com.br (M.d.F.N.d.S.A.); cassiavitoriaalves@gmail.com (C.V.d.D.A.S.); 5Instituto de Ciências Biológicais, Universidade Federal do Tocantins, Tocantins 77001-090, Brazil; 6Laboratório Central de Saúde Pública do Tocantins (LACEN/TO), Tocantins 77016-330, Brazil; 7Faculdade de Medicina do ABC, Santo André 09060-870, Brazil; skomninakis@yahoo.com.br; 8Laboratório de Retrovirologia, Universidade Federal de São Paulo, São Paulo 04023-062, Brazil; 9Instituto de Medicina Tropical, Universidade de São Paulo, São Paulo 05403-000, Brazil; degill@g.clemson.edu (D.E.G.); sabinoec@gmail.com (E.C.S.); charlysbr@yahoo.com.br (A.C.d.C.); 10Department of Obstetrics and Gynecology, Weill Cornell Medicine, 1300 York Avenue New York, NY 10065, USA; switkin@med.cornell.edu; 11Blood Systems Research Institute, San Francisco, CA 94143, USA; xdeng@bloodsystems.org; 12Department Laboratory Medicine, University of California San Francisco, San Francisco, CA 94143, USA; eric.delwart@ucsf.edu

**Keywords:** parechovirus, picornavirus, virome, recombination, coalescent, Brazil

## Abstract

We characterized the 24 nearly full-length genomes of human parechoviruses (PeV) from children in the north of Brazil. The initial phylogenetic analysis indicated that 17 strains belonged to genotype 1, 5 to genotype 4, and 1 to genotype 17. A more detailed analysis revealed a high frequency of recombinant strains (58%): A total of 14 of our PeV-As were chimeric, with four distinct recombination patterns identified. Five strains were composed of genotypes 1 and 5 (Rec1/5); five strains shared a complex mosaic pattern formed by genotypes 4, 5, and 17 (Rec4/17/5); two strains were composed of genotypes 1 and 17 (Rec1/17); and two strains were composed of genotype 1 and an undetermined strain (Rec1/und). Coalescent analysis based on the *Vp1* gene, which is free of recombination, indicated that the recombinant strains most likely arose in this region approximately 30 years ago. They are present in high frequencies and are circulating in different small and isolated cities in the state of Tocantins. Further studies will be needed to establish whether the detected recombinant strains have been replacing parental strains or if they are co-circulating in distinct frequencies in Tocantins.

## 1. Introduction

The *Picornaviridae* family, whose natural hosts are vertebrates [1,2,3], is currently divided into 47 genera (http://www.picornaviridae.com). *Enteroviruses, Hepatoviruses, Aphthoviruses*, and *Parechoviruses* are the most exhaustively characterized pathogens within this family and are associated with a wide range of clinical conditions, including respiratory disorders, gastroenteritis, myocarditis, sepsis, aseptic meningitis, encephalitis, and acute flaccid paralysis [2,4]. The *Parechovirus* genus includes species that infect humans, such as human parechovirus (PeV-A) and the zoonotic Ljungan virus (PeV-B). PeV-As were first isolated in 1956 and initially classified as enteroviruses (i.e., echoviruses 22 and 23) before being reclassified into the genus *Parechovirus* in 1997 [5,6,7]. PeV-A is transmitted by the fecal-oral route [8,9]. Serological studies indicated high levels of PeV-A infection in young children, mainly by the time they reach school age [8,10]. Although PeV-A infections have been associated with gastroenteritis, most are thought to be asymptomatic or cause only mild illness [4,9,11,12,13]. Notably, PeV-A of genotype 3 has been associated with severe infections in neonates [4]. PeV-A genomes consist of a positive sense RNA of approximately 7300 nucleotides, comprising a single open reading frame that encodes a polyprotein that is cleaved by viral proteases into structural and nonstructural proteins. Phylogenetic analysis, based on the VP1 gene, identified at least 19 different genotypes distributed globally [3,10,14,15]. In addition, full length genome analysis revealed a high frequency of various chimerical strains composed of both intra- and inter-genotype recombination [3,13,16,17]. Some studies identified breakpoint clustering around the junction between the capsid and nonstructural genes [3,16]. This pattern of recombination appears to be common in many members of the *Picornaviridae* family [18,19].

We report PeV-A detection in 24 children with acute gastroenteritis in a rural area in northern Brazil. Our results showed the occurrence of distinct PeV-A genotypes and a high frequency of recombinant strains.

## 2. Materials and Methods

### 2.1. Patients

This work was part of surveillance program carried out in the state of Tocantins, north Brazil, from 2010 to 2016. A total of 238 fecal specimens, collected between 2010 and 2016, were screened for enteric pathogens (i.e., rotavirus and norovirus), bacteria (i.e., *Escherichia coli* and *Salmonella sp*.), endoparasites (i.e., *Giardia sp.*), and helminthes, using conventional culture techniques and commercial enzyme immunoassays, such as RotaScreenII® and AdenoScreen® EIA (Microgen Bioproducts Ltd, 1, Watchmoor Point, Watchmoor Rd, Camberley GU15 3AD, UK). Subjects ranging from 0.5 to 2.5 years old, with the exception one 50 year old individual, suffered from acute gastroenteritis at the time of sampling, the signed informed consent was obtained. To identify possible undetected enteric viruses, NGS techniques were applied to all 238 samples using the method described below. Rotaviruses (*n* = 112), adenoviruses (*n* = 44), norovirus (*n* = 39), astroviruses (*n* = 8), and sapovirus (*n* = 8) were identified in some of these subjects.

### 2.2. Sample Processing

The protocol used to perform deep-sequencing was a combination of several protocols normally applied to viral metagenomics and/or virus discovery and has been previously described [20,21]. Briefly, 50 mg of the human fecal sample was diluted in 500 μL of Hank’s buffered salt solution (HBSS), added to a 2 mL impact-resistant tube containing lysing matrix C (MP Biomedicals, Santa Ana, CA, USA), and homogenized in a FastPrep-24 5G Homogenizer (MP biomedicals, USA). The homogenized sample was centrifuged at 12,000× *g* for 10 min and approximately 300 μL of the supernatant was then percolated through a 0.45 μm filter (Merck Millipore, Billerica, MA, USA) to remove eukaryotic- and bacterial-cell-sized particles. Approximately 100 μL, roughly equivalent to one fourth of the volume of the tube, of cold PEG-it Virus Precipitation Solution (System Biosciences, Palo Alto, CA, USA) was added to the filtrate and the contents of the tube were gently mixed and then incubated at 4 °C for 24 h. After the incubation period, the mixture was centrifuged at 10,000× *g* for 30 min at 4 °C. Following centrifugation, the supernatant (~350 μL) was discarded. The pellet, rich in viral particles, was treated with a combination of nuclease enzymes (TURBO DNase and RNase Cocktail Enzyme Mix-Thermo Fischer Scientific, Waltham, MA, USA; Baseline-ZERO DNase-Epicentre, Madison, WI, USA; Benzonase-Darmstadt, Darmstadt, Germany; and RQ1 RNase-Free DNase and RNase A Solution-Promega, Madison, WI, USA) to digest unprotected nucleic acids. The resulting mixture was subsequently incubated at 37 °C for 2 h. After incubation, viral nucleic acids were extracted using a ZR & ZR-96 Viral DNA/RNA Kit (Zymo Research, Irvine, CA, USA) according to the manufacturer’s instructions. The cDNA synthesis was performed with an AMV reverse transcription reagent (Promega, Madison, WI, USA). A second strand cDNA synthesis was performed using a DNA Polymerase I Large (Klenow) Fragment (Promega). Subsequently, a Nextera XT Sample Preparation Kit (Illumina, San Diego, CA, USA) was used to construct a DNA library, which was identified using dual barcodes. For the size range, Pippin Prep (Sage Science, Inc., Beverly, MA, USA) was used to select a 300 bp insert (range 200–400 bp). The library was deep-sequenced using a Hi-Seq 2500 Sequencer (Illumina, CA, USA) with 126 bp ends. Bioinformatics analysis was performed according to the protocol previously described by Deng et al [22]. The contigs, including sequences of rotaviruses, as well as enteric viruses, humans, fungi, bacteria, and others, sharing a percent nucleotide identity of 95% or less were assembled from the obtained sequence reads by de novo assembly. The resulting singlets and contigs were analyzed using BLASTx to search for similarity to viral proteins in GenBank. The contigs were compared to the GenBank non-redundant nucleotide and protein databases (BLASTn and BLASTx). After identification of the viruses, a reference template sequence was used for mapping the full-length genome with Geneious R9 software (Biomatters Ltd L2, 18 Shortland Street Auckland, 1010, New Zealand). Based on the best hits of the BLASTx searches, PeV-A genomes were chosen for further analyses. Sequences generated in this study have been deposited in GenBank. Accession numbers are MK904585–MK904608.

### 2.3. Alignment and Phylogenetic Analysis

These genomes were then aligned using Clustal X software [23]. Subsequently, a phylogenetic tree was constructed using the maximum likelihood approach and branch support values were assessed using the Shimodaira–Hasegawa test. All trees were inferred using FastTree software [24]. The GTR model and gamma distribution were selected according to the likelihood ratio test (LRT) implemented in the jModeltest software [25]. Likelihood mapping was obtained using the software Tree-puzzle, version 5.3 [26], assuming the GTR model and rate of heterogeneity for the evolutionary model. Analyses were performed using 1000 replications.

### 2.4. Detection of Recombination

We used RDP v.4 software [27], which utilizes a collection of methods, to determine the extent of recombination among sequences. Below is a brief description of these methods and an excellent and detailed explanation of each method implemented in the RDP program can be found in the user’s manual (http://darwin.uvigo.es/rdp/rdp.html). Maximum χ2 (MaxChi) is a method, implemented by Maynard Smith, which uses variable/invariable sites to detect recombination in pairs of sequences. This method generates random sequence pairs. The significance level is evaluated by the proportion of simulated sequence pairs with maximum χ2 values higher than the real data. The maximum match χ2 (Chimaera) is a modification of Smith’s method. It uses variable sites to calculate the maximum χ2 match statistics. Geneconv detects gene conversions (recombination) by evaluating conserved substitutions in fragments between two sequences. Although evolutionary methods are not explicitly implemented in Geneconv, it is robust and has low levels of false positive detection of recombination, including those events due to rate heterogeneity and natural selection. Bootscanning is a sliding window method that was developed to identify the parental origins of sequence fragments (windows) within known or putative recombinant sequences. Lard is similar to MaxChi and the method scans an alignment of three sequences (a recombinant and two parental sequences) for the point in the alignment that optimally separates regions of conflicting phylogenetic signals; *p*-values are also estimated to the breakpoint. Initially, we used default parameters. We later optimized the parameters to avoid detection of false positive recombination. In addition, window sizes of 50 to 350, stepping of 50–100 nt, as well as Bonferroni correction with *p*-values of 0.05 and 0.001 were utilized.

### 2.5. Coalescent Analysis

We used a Bayesian Markov chain Monte Carlo (BMCMC) coalescent framework to estimate the ancestral genealogy and evolutionary parameters, such as nucleotide substitution rates per year and time to the most common ancestor (tMRCA). Strict and relaxed molecular clocks were evaluated and the GTR model, plus a gamma correction, was applied to all analyses. The evolutionary and demographic parameters were iteratively adjusted. We used constant population size coalescent prior models to determine the dates of clades of PeV-A in Brazil. We also used the Bayesian skyline plot method [28] (BSP), which provides an unbiased description of genetic diversity over time (expressed as the product of the effective sample size-Ne and generation time-τ). Since we used absolute time (years) to scale branch lengths and did not assume a specific generation time, our estimates of *Ne.τ* will reflect only the relative genetic diversity of infections over time. In our analyses, 122 sequences (described in Appendix A) sampled at different times (heterochronous) were used to estimate the ancestral genealogy, evolutionary parameters such as nucleotide substitution rates per year, and diversity over time. The Markov chain Monte Carlo (MCMC) processes were run for 250,000,000 generations, with the initial 10% of each run discarded as burn-in. We used a GTR model and gamma correction to accommodate rate heterogeneity. Ambiguously aligned sites were removed. The selection of coalescent models and molecular clocks were performed by Bayes factor testing on marginal likelihood estimation, obtained using path sampling and the stepping-stone sampling approach [29]. Constant population size, exponential growth, and Bayesian skyline models were evaluated besides the strict molecular clock and the relaxed molecular clock, which allow rates to vary among lineages. The lognormal molecular clock yielded the highest likelihood in all coalescent models. Relaxed clocks and rate heterogeneity are usually used to reduce the effect of unequal mutation rates among lineages and also to avoid bias of purifying selection [30,31]. To estimate the mutation rate, we assumed 2 × 10^−3^ site substitutions per year (s/y). We also used prior probability distributions, such as a lognormal distribution, for the sample size with a mean of 300 ± 100 and lognormal distributions for the clock rate with a mean of 2.7 × 10^−3^ ± 0.001. Mutation rates were assumed based on previous estimates of PeV-A evolution [30]. Lognormal distributions were incorporated to avoid bias in the branch lengths caused, for instance, by a uniform prior distribution [31,32]. Thus, new estimates, based on the best-fitted relaxed lognormal molecular clock models, for substitution rates and their 95% highest posterior density (HPD) interval (the credible interval that contains 95% of all the samples) were obtained. The convergence of chains was evaluated using TRACER software, version 1.7.1 [33]. Runs were accepted when all parameters presented an effective sample size number, (ESS) > 200. Two independent chains were run for each dataset and combined with LogCombiner software. All these analyses were performed with the BMCMC approaches, implemented in the BEAST package version 1.10.4 [34].

## 3. Results

Following the initial screening, next generation sequencing techniques were used to identify the possible presence of additional enteric viruses. Viral sequences were identified through sequence identity (using Blast) to annotated viral genomes in GenBank. After identification of PeV-As, the closest matching VP1 sequences were used to assign genotypes. The map in Figure 1 shows where PeV-As were detected and the numbers of each genotype based on the phylogenetic tree inferred from the genome (see below).

### 3.1. Genotyping Tree

The near full-length genomes (approximately 7400 nucleotides) of our Brazilian PeV-As strains were sequenced and compared with 98 previously described near full-length parechovirus genome strains. The phylogenetic tree constructed using the genome is shown in Figure 2. This maximum likelihood tree shows that sequences of genotypes 1, 2, 3, 4, 5, and 6 (PeV-A1, PeV-A2, PeV-A3, PeV-A4, PeV-A5, and PeV-A6) cluster in independent clades, all having high values of branch support (approximate likelihood ratio test-aLRT). These clades correspond to the genotypes PeV-A1, PeV-A2, PeV-A3, PeV-A4, PeV-A5, and PeV-A6, which were previously determined by Zell et al., 2017 [2]. It also demonstrates that eighteen of our strains (i.e., #09, 21, 43, 47, 48, 50, 53, 54, 65, 66, 82, 88, 110, 158, 170, 175, 183, and 237) are located within the clade of PeV-A genotype 1. Another strain (#102) is related to the PeV-A reference of genotype 17 and five (17, 23, 74, 80, and 84) fall within the clade formed by genotype 4 strains. The tree also has 26% of star-trees, as was determined by the likelihood mapping approach (central area in the triangle in Figure 2). This approach shows the percentage of unresolved star-trees in the alignment in the center of the triangle. The higher this percentage, the more inferior is the alignment for phylogenetic inferences. Note that some groups were collapsed to facilitate visualization of the tree. For comparative purposes, we also constructed trees using the VP1 and 3D proteins (see Appendix A). The VP1 tree has 39% of star-trees and is nearly identical to the full genome nucleotide tree. In addition to the same pattern of topology, the genotyping of our sequences was the same. The tree inferred using the gene *3D* has 47% of star-trees, low branch support, and there is no clustering pattern of strains.

### 3.2. Recombination Analysis

We proceeded to identify chimeric PeV-A genomes. Initially, 98 reference sequences were analyzed to identify non-recombinant strains to be used as “parental” strains. The strains KT879919 (PeV-A2); KT879918 (PeV-A5); FJ888592 (PeV-A6); JX682576 (PeV-A3); KY404171 (PeV-A4); and KT879922 (PeV-A1) were used as parental references in the following analysis. We also excluded strains #021, #065, and #175 since they were of poor quality and could reduce reliability of the analysis. Next, we screened a dataset composed of the parental references and Brazilian PeV-A strains. In addition to the strains generated in this study, the alignment used also contained other Brazilian strains, as follows: EU716175 (PeV-A8-BR/217); HQ696577 (PeV-A6-BR/104); HQ696576 (PeV-A5-BR/77); HQ696575(PeV-A5-BR/53); HQ696572 (PeV-A1-BR/30) HQ696571 (PeV-A1-BR/27); HQ696574 (PeV-A1-BR/145); HQ696570 (PeV-A1-BR/21); and HQ696573 (PeV-A1-BR/114). This allowed us to detect many recombinant PeV-As (i.e., 17, 23, 43, 47, 50, 53, 54, 74, 80, 84, 110, 158, 183, and 237) among the sequences generated. These recombinant strains are fully characterized in the next section.

### 3.3. Mosaic Pattern

We also determined the mosaic pattern and position of the recombination breakpoint of the Brazilian recombinant strains. Only the polyprotein region was used to accomplish this because most sequences presented large indels in the 5’ and 3’ untranslated regions. Initially, we performed screening using methods (i.e., GeneConv, MaxChi, BootScan, and Lard) that combined different approaches to detect recombination in the alignment. Based on this methodology, we were able to identify four distinct categories of chimeric genomes. Figure 3 is based on bootscan analysis and describes the mosaic patterns of the categories of recombinant strains that we found. The strain Rec1/und. is represented by sequences (i.e., 47, 50) that have genomes composed of genotypes 1 and an undetected genotype. They have a single breakpoint located at position 5273 of the polyprotein. Another similar pattern, composed of genotypes 1 and 17, is the recombinant named Rec 1/17, represented by strains 110 and 183. They have a breakpoint at position 4736 of the polyprotein. The category named Rec4/17/5 is a complex chimera composed of genotypes 4, 5, and 17. There are five strains in this category (i.e., 17, 23, 74, 80, and 84) and breakpoints are located in positions 2009, 3032, and 4048 of the polyprotein. Lastly, the category of recombinants named Rec1/5 is composed of strains 43, 53, 54, 158, and 237 and has a single breakpoint at position 3613 in the polyprotein, which makes this a chimera between genotypes 1 and 5. Interestingly, all strains of the category Rec4/17/5 are monophyletic in the tree and inferred using gene *Vp1* or the capsid genes *Vp0*, *Vp3*, and *Vp1*. In addition, Rec 1/17 plus Rec1/und are monophyletic in the *Vp1* and capsid genes trees. Likewise, strains of the category Rec 1/5 are also monophyletic in the *Vp1* and capsid genes trees. We also found that some previously reported PeV-As are also recombinant strains (see Appendix A).

### 3.4. Dynamics of PeV-A Infection

To assess dynamics of PeV-A infection, we used an alignment corresponding to the *Vp1* gene composed of 122 PeV-A sequences included in 30 Brazilian strains. In addition to our samples, we also used strains of PeV-A identified in 2006 from the northeast state of Bahia [17], which is adjacent to Tocantins (see map in Figure 1). Previously, we performed a detailed analysis showing no signal of recombination in this genomic region (VP1) among Brazilian samples. Table 1 shows the statistical parameters calculated in each model. The constant size is the coalescent model with the highest marginal likelihood. Substitution rates of 2.2 × 10^−3^ (95% HPD = 1.6 × 10^−3^–2.7 × 10^−3^) and tMRCA of 1552 (95% HPD = 1270–1760) were obtained. These values are in agreement with previous estimates performed using the *Vp1* gene of PeV-A [30].

### 3.5. Timescale for the PeV-A Recombinant Strains in Brazil

The maximum clade credibility (MCC) tree, inferred using the recombination-free *Vp1* gene and the constant size model, was used to compare dates of Brazilian recombinant strains and parental strains (Figure 4). The tree also shows that nodes of the clusters containing recombinant strains (indicated by green areas in the MCC tree) are younger than nodes of parental strains. For example, the recombinant strain Rec4/17/5 diverged from parental strains around 1987 (95% HPD interval of 1969–1999). Similarly, the recombinant strains Rec1/ diverged from the parental strains around 1994 (95% HPD interval of 1984–2001).

## 4. Discussion

In our survey of 238 individuals affected by acute gastroenteritis, we found PeV-A in 24 patients. Most of them were also infected with other viruses, mainly rotavirus (*n* = 112). For this reason a causative link between PeV-A and diarrhea in our study population is uncertain. In addition, our survey suggests that PeV-A is endemic in this region of Brazil, owing to the large PeV-A genomic diversity observed in small and isolated cities. The nearly complete genomes of these 24 PeV-A strains were sequenced and compared with other strains previously identified worldwide. Phylogenetic analysis revealed the presence of PeV-A genotypes 1, 4, and 17 in the region. Previous studies indicated the occurrence of genotype 1, 5, 6, and 8 in the state of Bahia, but not genotypes 4 and 17 [17,35]. In agreement with previous studies, phylogenies inferred from the near full-length genome and capsid genes also showed that strains of genotype 1 form two divergent clades supported by high aLRT values (indicated by stars in Figure 2). These clades (PeV-A1a and PeV-A1b) have been observed in previous studies [15]. They are not artifacts of recombination because the segregation of genotype 1 is equally observed in trees inferred using regions free of recombination. Furthermore, the split of genotype 1 is likely old because strains from distinct world regions are present in both clades (Figure 2 and Appendix A). Indeed, our Bayesian coalescent analysis suggested that clades PeV-A1a and PeV-APeV-A1b diverged from each other at least 96 years ago. There are some Brazilian strains clustering within clade PeV-A1a and other strains clustering within clade PeV-A1b. Another relevant feature shown by phylogenetic analysis is that the Brazilian PeV-A1 strains are not monophyletic. This is probably a consequence of the initial introduction of multiple strains in these regions, followed by rapid adaptive mutations in the local host population.

Recombination in PeV and other single stranded RNA viruses has been previously discussed and many studies show that breakpoints are typically located between the capsid and non-structural genes [3,16,36]. We have now identified strains formed by the recombination of distinct genotypes. Particularly, PeV-A infections in Tocantins are characterized by recurrent recombination events between parental genotypes that were not previously reported in Brazil. These recombinants were classified into four distinct lineages and were named according to the parental genotypes that formed their genomes (i.e., Rec1/5; Rec4/17/5; Rec1/17, and Rec1/und.). The most significant finding was the high frequency of individuals infected with certain recombinant strains, indicating their successful spread. Particularly, recombinant strains Rec1/5 and Rec4/17/5 were most frequently detected among our sampled subjects. It is important to mention that the parental lineage strains that composed the genomes of Rec1/5 and Rec4/17/5 were absent or present in low frequencies in the region. Taking into account that the chimeric strains were found in different rural and isolated areas, it is reasonable to suggest that recombinant PeV-As are endemic. The tMRCA of Brazilian strains was estimated based on the *Vp1* gene region, which is free of recombination, and shows that these PeV-A genotypes originated in distinct time scales. For example, PeV-A1a diverged in 1879 (1835–1920) and PeV-A1b diverged more recently in 1955 (1924–1977). Our study showed that, in Tocantins, there are multiple genotypes circulating with distinct frequencies. Some of these genotypes were likely introduced during the first colonization period while others likely appear to have reached the region more recently. The first settlements in the Araguaína region were in 1876. This region was extremely isolated from other municipalities until 1925, owing to the lack of roads. Afterwards, many families migrated there from other states, mainly from the northeast regions of Brazil. The PeV-A diversity of distinct genotypes in Tocantins is possibly the consequence of the migratory influx of people from distinct regions. Indeed, our analysis suggests that recombinant strains have arisen recently, approximately 30 years ago, and appear to be efficiently maintained, as evidenced by many individuals being infected by these strains. The rapid dissemination of nearly identical recombinant strains, along with the absence or low frequency of parental lineages, suggests that recombinant PeV-A strains are endemic in the region of the study.

In summary, we describe that PeV-A in remote areas in the north of Brazil exhibits a great diversity of genotypes and common recombination forms, which have recently spread within the host population.

## Figures and Tables

**Figure 1 viruses-11-00488-f001:**
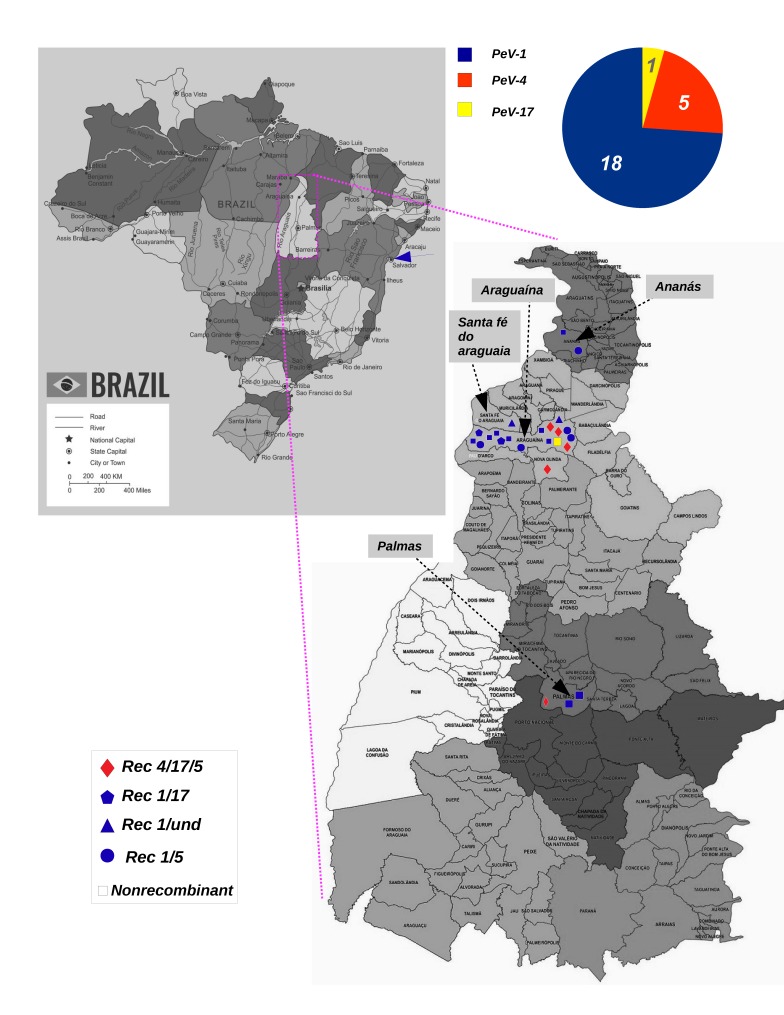
Location of PeV-A strains detected in Brazil. The map of Brazil is shown at the top of the figure to illustrate the region (magenta rectangle) in Tocantins state where samples were collected. A diagram shows the proportions of genotypes detected in this study (top-right). The magnified image of the region shows the cities (indicated by arrows) and the location of each PeV-A isolate. Distinct recombinant strains are represented by different shapes, as is indicated in the panel (bottom-left). Colors indicate distinct PeV-A genotypes. The blue arrows indicate the city (Salvador) where PeV-A strains were detected previously in Brazil.

**Figure 2 viruses-11-00488-f002:**
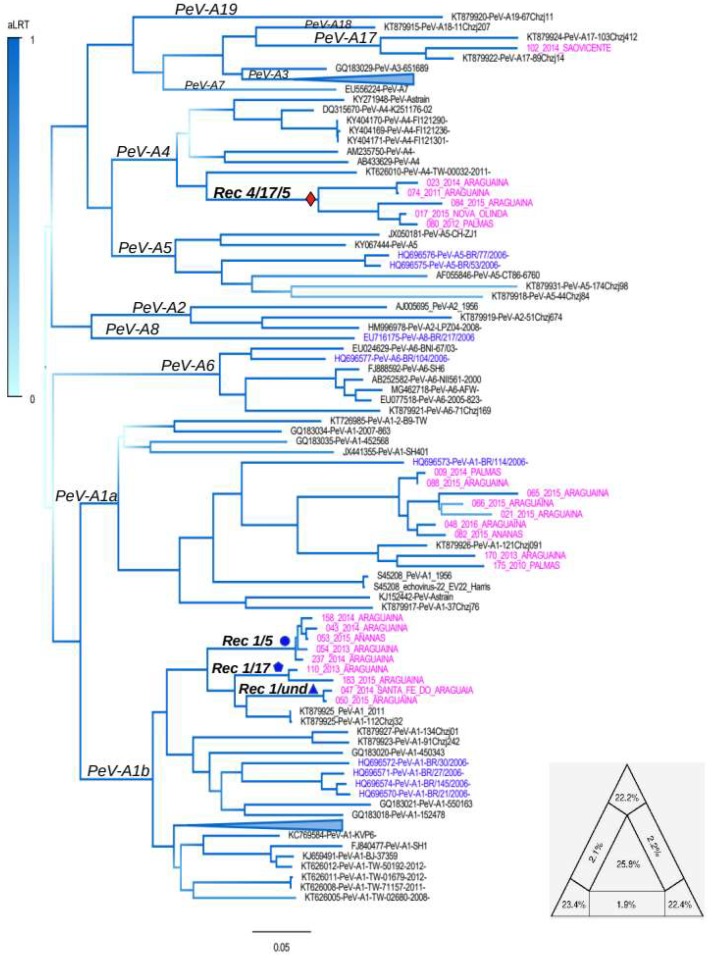
Maximum likelihood tree constructed using near-full length genome of PeV-A. The Brazilian strains described in the present study (diamonds) and from previous studies (arrows) are indicated in the tree. A colored scale indicating the statistical support of each node, calculated using aLRT, is shown in the tree. Phylogenetic groups corresponding to main genotypes (only the genotypes in which full length genomes were available) are indicated. The scale bar under the tree represents the nucleotide substitutions per site. |A maximum likelihood tree was inferred assuming the GTR+gamma model and was constructed using the software FastTree [24]. The triangle in the base of the tree is the likelihood map and it shows 25.9% of unresolved trees in the PeV-A genome alignment. Likelihood mapping was obtained using the software Tree-puzzle, version 5.3 [26], assuming the GTR model and the rate of heterogeneity for the evolutionary model. Analyses were performed using 1000 replications.

**Figure 3 viruses-11-00488-f003:**
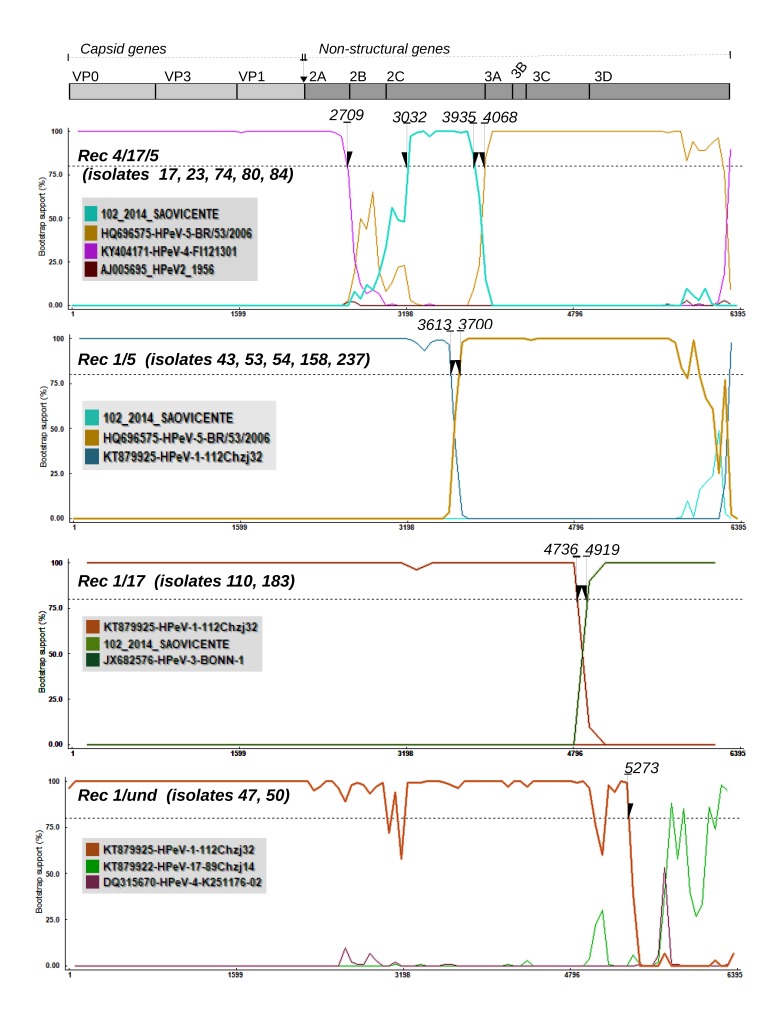
Recombination pattern of chimera strains of PeV-A from regions in northern Brazil. The bootscanning method was used to determine the parental genotypes that compose the recombinant PeV-A strain. Colored lines represent the probability (given in bootstrap value) of genomic regions belonging to a certain parental genotype. The x-axis represents the sequence length in base pairs (bp). The y-axis represents the statistical support (bootstrap) based on 500 replicates. In the upper region of the figure, a diagram shows the genome map of PeV-A. Each plotted line refers to a certain genotype (see the sequence code color in the gray panel). Each plot indicates the breakpoints in the polyprotein region of the following four categories of PeV-A recombinants: Rec1/5, Rec1/17, Rec1/und, and Rec4/17/5. Isolates that belong to a certain recombinant category are listed within parenthesis. The evolutionary model (Felsenstein, 1984) plus the estimated transition/transversions (ts/tv = 2.7) were used. Window sizes of 100 to 650, stepping of 60–150 nt, as well as χ2 correction, with *p*-values of 0.05 and 0.001, were utilized. All these analyses were performed using the RDP v4 software [27].

**Figure 4 viruses-11-00488-f004:**
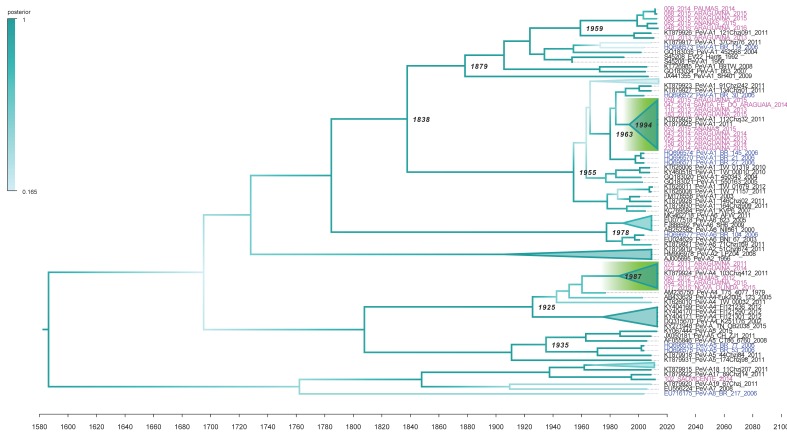
Maximum clade credibility tree based on the *Vp1* gene of PeV-A. The Bayesian time-scaled tree maximum clade credibility tree was inferred using the relaxed molecular clock and the constant population size model. The x-axis represents chronological time, expressed in years. Sequences generated in the current study are indicated in magenta and Brazilian strains from a previous study are indicated in blue. Clades containing recombinant strains described in this study are indicated by green areas in the tree. The tree also shows (numbers at nodes) the divergence times of lineages, expressed in mean posterior estimate, of ages calibrated in years from the tMRCA. The branch color indicates the posterior probability as indicated by the scale in the left region of the figure. The main genotypes are indicated above the branches. PeV-A3 strains were collapsed in the tree. The tree was summarized using TreeAnnotator software. All these analyses were performed with the BMCMC approaches implemented in the BEAST package, version 1.10.4 [34].

**Table 1 viruses-11-00488-t001:** Statistical parameters obtained under the relaxed lognormal molecular clock.

*Vp1 gene* (*n* = 122, l = 556)			
Coalescent Model	MLE	Substitution Rates *	tMRCA *
Constant size	−16,618.29	2.2 × 10^−3^(1.6 × 10^−3^–2.7 × 10^−3^)	1552 (1270–1760)
Exponential growth	−16,696.69	1.8 × 10^−3^(1.1 × 10–2.5 × 10^−3^)	1489 (1143–1747)
BSL	−17,212.22	1.7 × 10^−3^(1.0 × 10^−3^–2.5 × 10^−3^)	1482 (1248–1780)

BSL = Bayesian skyline, MLE = Marginal likelihood estimates. * 95% lower and. upper bounds of the highest probability density intervals in parentheses.

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
