# Peer review of "Recombinant Strains of Human Parechovirus in Rural Areas in the North of Brazil"

_viruses, 2019, doi:10.3390/v11060488_

Round 1
Reviewer 1 Report
The manuscript by Leal et al entitled ‘Recombinant strains of human parechovirus in rural areas in the North of Brazil’ presents the identification of HPeV genomes in the 24 (out of 238) clinical samples of young patients suffering from acute gastroenteritis as well as phylogenetic analysis of the identified genomes. The identified genotypes were assigned to HPeV1, HPeV4 and HPeV17; more than half of the HPeV genomes were recombinants. The recombination occurred approximately 30 years ago in the region. Authors suggest that recombinant strains are endemic in the region. Furthermore, authors identify that Brazilian HPeV1 strains are not monophyletic indicating that HPeV1 was introduced more than once to the region. This is important information for monitoring virus spread, adaptation and pathogenesis.
Manuscript describes HPeV genotypes (HPeV4 and HPeV17) that were not identified before in Brazil. Though authors identified a number of recombinant HPeV strains circulating in the region of Brazil, the clinical significance of these strains is still unclear. This could be due to a relatively small size of the clinical sample collection (238 samples) originating from a small region of the country as well as age of the patients (3 – 14 years old), as it is known that HPeV3 causes most severe infections in neonates.
Specific comments are below.
line 50: currently Picornaviridae family consists of 110 species grouped into 47 genera (as of February 2019) (from http://www.picornaviridae.com). The information in the manuscript should be updated. In addition, poliovirus is not a genus and not a species; polioviruses belong to the Enterovirus genus, Enterovirus C species. If authors refer to genera, then it should be Enterovirus, Hepatovirus etc. If authors mention isolates, those should be listed as enteroviruses, polioviruses etc.
Lines 50-54: As viruses from Picornaviridae infect vertebrate animals, you could leave ‘animal’ out in the sentence: ‘Enteroviruses, polioviruses, hepatoviruses, aphthoviruses and parechoviruses are the most exhaustively characterized pathogens within this family and are associated with a wide range of clinical conditions, including respiratory disorders, gastroenteritis, myocarditis, sepsis, aseptic meningitis, encephalitis and acute flaccid paralysis (Zell et al., 2017).’
Line 54: authors should be more precise with listing parechovirus species that can infect humans, therefore ‘some' should be avoided in this sentence: ‘The Parechovirus genus includes some species that infect humans, such as human parechovirus (HPeV) and the zoonotic Ljungan virus.’
Lines 60-62: authors miss general information on HPeV3, which can cause severe infections in neonates (e.g. Olijve et al., 2018).
Lines 72-76 authors write: ‘However, the role of recombination in the evolution of new lineages of HPeV has to date not been addressed.
We report here HPeV detection in 24 children with acute gastroenteritis in a rural area in Northern Brazil. Based on the high diversity of genotypes and recombinant strains identified, HPeV infections are likely endemic in this region.’ Based on the first sentence, readers expect that authors will explain the role of recombination in evolution of new HPeV lineages. The following sentences do not present that. This part needs more elaboration (rephrasing).
Lines 90-92: authors need to provide a brief description for the protocol used.
Line 139: It is not very clear how authors chose the number for mutation rate and is it given per nucleotide, per genome, etc. Explanation or reference would help the reader.
Line 166: Is it possible to indicate the shortest and the longest sequence of the genomes used?
Lines 186 – 187: no need to introduce the abbreviation once again.
Lines 213-215: Not clear sentence ‘Initially, we performed screening using a some method (i.e., GeneConv, MaxChi, BootScan and Lard) that combined different approaches to detect recombination in the alignment.’ Statement that is more precise is needed here. Please reword.
Lines 229-230: sentence ‘We also found some that some HPeV previously reported are also recombinant strains’ has too many ‘some’. Needs fixing.
Lines 246-247: It is not clear what data set was used in this analysis.
Lines 249-250: It is not clear where the previous analysis was published.
Figure 1: The labels in the figure are not legible. The inset on the Brazil map as well as labels for cities etc should be enlarged to make them readable. Authors could consider excluding some labels and leaving only most important for orientation. The design of the figure is not optimal. The pie diagram on top right would better fit to right down corner and the map for Tocantins region aligned up with Brazil map. The boxed part on the Brazil map is not seen, it would be better to color the region in a bright color like pink or green etc. Bottom left corner, nonrecombinant box should be shown in yellow. On the Brazil map, it is worth considering to mark where other HPeV isolates were collected (published in Drexler et al 2011).
Figure 2: The labels for HPeV isolates are not readable as they partially overlap. The font or font size should be changed.
Figure 3: Labels on the axes are too small. The labels in grey insets are fuzzy. Resolution should be increased.
Figure 4: Labels on the axis are too small; numbers at nodes could be bigger; the scale is not readable; the labels for the isolates are not readable and overlapping. The figure design should be changed to make fonts readable. Authors could consider to present this figure in landscape position, if journal allows.
Suppl figures S1 and S2: The labels for the isolates are overlapping or too close, needs fixing.
Typing errors like on line 170 (HPeV6 instead of HpeV6 and independent instead of independete) should be corrected.
Author Response
The manuscript by Leal et al entitled ‘Recombinant strains of human parechovirus in rural areas in the North of Brazil’ presents the identification of HPeV genomes in the 24 (out of 238) clinical samples of young patients suffering from acute gastroenteritis as well as phylogenetic analysis of the identified genomes. The identified genotypes were assigned to HPeV1, HPeV4 and HPeV17; more than half of the HPeV genomes were recombinants. The recombination occurred approximately 30 years ago in the region. Authors suggest that recombinant strains are endemic in the region. Furthermore, authors identify that Brazilian HPeV1 strains are not monophyletic indicating that HPeV1 was introduced more than once to the region. This is important information for monitoring virus spread, adaptation and pathogenesis.
Manuscript describes HPeV genotypes (HPeV4 and HPeV17) that were not identified before in Brazil. Though authors identified a number of recombinant HPeV strains circulating in the region of Brazil, the clinical significance of these strains is still unclear. This could be due to a relatively small size of the clinical sample collection (238 samples) originating from a small region of the country as well as age of the patients (3 – 14 years old), as it is known that HPeV3 causes most severe infections in neonates.
Specific comments are below.
line 50: currently Picornaviridae family consists of 110 species grouped into 47 genera (as of February 2019) (from http://www.picornaviridae.com). The information in the manuscript should be updated. In addition, poliovirus is not a genus and not a species; polioviruses belong to the Enterovirus genus, Enterovirus C species. If authors refer to genera, then it should be Enterovirus, Hepatovirus etc. If authors mention isolates, those should be listed as enteroviruses, polioviruses etc.
Resp; We have updated the picornavirus classification and corrected the nomenclature of theses viruses in the new version of the manuscript.
Lines 50-54: As viruses from Picornaviridae infect vertebrate animals, you could leave ‘animal’ out in the sentence: ‘Enteroviruses, polioviruses, hepatoviruses, aphthoviruses and parechoviruses are the most exhaustively characterized pathogens within this family and are associated with a wide range of clinical conditions, including respiratory disorders, gastroenteritis, myocarditis, sepsis, aseptic meningitis, encephalitis and acute flaccid paralysis (Zell et al., 2017).’
Resp; “animal” removed from the sentence.
Line 54: authors should be more precise with listing parechovirus species that can infect humans, therefore ‘some' should be avoided in this sentence: ‘The Parechovirus genus includes some species that infect humans, such as human parechovirus (HPeV) and the zoonotic Ljungan virus.’
Resp; We agree with the referee in this point and excluded the word “some” from this sentence.
Lines 60-62: authors miss general information on HPeV3, which can cause severe infections in neonates (e.g. Olijve et al., 2018).
Resp; We have included in the new manuscript the sentence “HPeV of genotype 3 has been associated with severe infections in neonates (Olijve et al., 2018).”Lines 57-58
Lines 72-76 authors write: ‘However, the role of recombination in the evolution of new lineages of HPeV has to date not been addressed.Resp this sentence was excluded to avoid misinterpretations
We report here HPeV detection in 24 children with acute gastroenteritis in a rural area in Northern Brazil. Based on the high diversity of genotypes and recombinant strains identified, HPeV infections are likely endemic in this region.’ Based on the first sentence, readers expect that authors will explain the role of recombination in evolution of new HPeV lineages. The following sentences do not present that. This part needs more elaboration (rephrasing).Resp; also changed the last sentence accordingly
Lines 90-92: authors need to provide a brief description for the protocol used.Resp; We briefly explained how samples were processed in this new version of the manuscript (Lines 88-111).
Line 139: It is not very clear how authors chose the number for mutation rate and is it given per nucleotide, per genome, etc. Explanation or reference would help the reader. Resp; Prior rates and distributions were assumed based on previous studies. The sentence “Mutations rates were assumed based on previous estimates of HPeV evolution (Faria et al., 2009). Lognormal distributions were incorporated to avoid bias in the branch lengths caused, for instance, by a uniform prior distribution (Yang and Rannala, 2005).” was included in the new manuscript.
Line 166: Is it possible to indicate the shortest and the longest sequence of the genomes used?Resp; The longest is isolate 080 (7408nt and the shortest is 065 (4107nt) that was excluded from most analysis
Lines 186 – 187: no need to introduce the abbreviation once again.Resp; This was changed
Lines 213-215: Not clear sentence ‘Initially, we performed screening using a some method (i.e., GeneConv, MaxChi, BootScan and Lard) that combined different approaches to detect recombination in the alignment.’ Statement that is more precise is needed here. Please reword. Resp; We have complemented the explanation with the following paragraph; To determine the extent of recombination among sequences, we used software RDP v.4 (Martin et al., 2015), which utilizes a collection of methods. Below is a brief description of these methods; an excellent and detailed explanation of each method implemented in the RDP program can be found in the user’s manual (http://darwin.uvigo.es/rdp/rdp.html). Maximum χ2 (MaxChi) is a method implemented by Maynard-Smith, and it uses variable/invariable sites to detect recombination in pairs of sequences. This method generates random sequence pairs; the significance level is evaluated by the proportion of simulated sequence pairs with maximum χ2 values higher than the real data. The maximum match χ2 (Chimaera) is a modification of Smith’s method. It uses variable sites to calculate the maximum χ2 match statistics. Geneconv detects gene conversions (recombination) by evaluating conserved substitutions in fragments between two sequences. Although evolutionary methods are not explicitly implemented in Geneconv, it is robust and has low levels of false positive detection of recombination, including those events due to rate heterogeneity and natural selection. Bootscanning is a sliding window method that was developed to identify the parental origins of sequence fragments (windows) within known or putative recombinant sequences. Lard is similar to MaxChi, and the method scans an alignment of three sequences (a recombinant and two parental sequences) for the point in the alignment that optimally separates regions of conflicting phylogenetic signals; p-values are also estimated to the breakpoint. Initially, we used default parameters; we later optimized the parameters to avoid detection of false positive recombination. In addition, window sizes of 50 to 350, stepping of 50-100nt, as well as Bonferroni correction with p-values of 0.05 and 0.001 were utilized.
Lines 229-230: sentence ‘We also found some that some HPeV previously reported are also recombinant strains’ has too many ‘some’. Needs fixing. Resp; Fixed
Lines 246-247: It is not clear what data set was used in this analysis.Resp; dataset is the alignment. We changed the sentence accordingly.
Lines 249-250: It is not clear where the previous analysis was published.Resp; We have cited the reference corresponding to these sequences (i.e., Drexlet et al., 2011)
Figure 1: The labels in the figure are not legible. The inset on the Brazil map as well as labels for cities etc should be enlarged to make them readable. Authors could consider excluding some labels and leaving only most important for orientation. The design of the figure is not optimal. The pie diagram on top right would better fit to right down corner and the map for Tocantins region aligned up with Brazil map. The boxed part on the Brazil map is not seen, it would be better to color the region in a bright color like pink or green etc. Bottom left corner, nonrecombinant box should be shown in yellow. On the Brazil map, it is worth considering to mark where other HPeV isolates were collected (published in Drexler et al 2011).Resp; Please check the original files of these figures. The resolution was increased and we also indicate in figure 1 the location of previous HPeV strains detected in Salvador, Bahia
Figure 2: The labels for HPeV isolates are not readable as they partially overlap. The font or font size should be changed.
Figure 3: Labels on the axes are too small. The labels in grey insets are fuzzy. Resolution should be increased.
Figure 4: Labels on the axis are too small; numbers at nodes could be bigger; the scale is not readable; the labels for the isolates are not readable and overlapping. The figure design should be changed to make fonts readable. Authors could consider to present this figure in landscape position, if journal allows.
Suppl figures S1 and S2: The labels for the isolates are overlapping or too close, needs fixing.
Typing errors like on line 170 (HPeV6 instead of HpeV6 and independent instead of independete) should be corrected. Resp; The new manuscript was revised and typos corrected.
Reviewer 2 Report
The authors describe genetic analysis of several parechoviruses, including recombinant strains. This study adds important information on parechovirus epidemiology in Brazil, as reports from South America are limited. It is a good paper and it should definitely be published after minor revisions. However, the writers should pay attention to details, which are described below.
Specific comments:
Check carefully the nomenclature of viruses and the use of italics in virus name? Species and genus name in italics, virus name not in italics. There are good instructions on the journal pages.
Human parechovirus species is now called parechovirus-A (PeV-A) and Ljungan virus is PeV-B. Also the proof that Ljungan virus can infect humans remains an open question. HPeV and LjV are still being used in publications, but I suggest using the new names. HPeV-1 => PeV-A1, HPeV-4 => PeV-A4 etc.
The authors should carefully go through the abbreviations: throughout the text human parechoviruses there are several different forms, HPeV or HpeV. These are also in figures (Fig 1).
The same is here with genome names: “VP1 gene region”, “Vp1 gene”, “gene VP1”, “Vp1”, all these can be found in the text as well as figure legends. (check also 3D)
In general, symbols for genes are italicized (e.g., IGF1), whereas symbols for proteins are not italicized (e.g., IGF1).
Figure 3 is not referred to in the text. Add reference.
Abstract:
The first sentence is a bit confusing. Persistence of a virus could mean also that the infection in an individual is persistent or chronic. Here it does not mean this. The first sentence could simply be removed.
There is possibly a typo: 24 strains (17 HPeV1; 5 HPeV4; 1 HPeV17, makes only 23). In figure 3 there are 18 HPeV1 types.
In addition, the number of specific types (18 HPeV1; 5 HPeV4; 1 HPeV17) is only given in figure 1 and the abstract. It should be added to results as well, in the sentences in rows 172-176 this could be specified.
Introduction:
Poliovirus is not a genera nor a species. Row 50: could be “Enteroviruses, especially (or including) polioviruses, Hepatoviruses …”
Materials and methods:
Genebank accession numbers should be included before publication.
Results:
Row 184: “strains that belong to the same VP1 genotype”. Should this be HPeV1 genotype?
Row 229-231 the last sentence of the chapter has the word “some” twice.
Discussion:
Rows 279-283: First two sentences should rather be included in the background, not here.
What could be mentioned here is that in this study 10% of these diarrhea samples were positive, which is quite high when compare to many other studies. Also, most HPeV positive stool samples are found from slightly younger children, below the age of 2 years. In younger children the detection rate would probably have been somewhat higher. This could be mentioned in discussion even though this is not the main thing in this study.
Supplementary figures:
there are two “Figure S2. 3D tree of HPeV”, but the actual figures look different from each other. The latter Should probably be S3?
First Figure S2, the color codes in the figure are different from what is said in the legend text.
Author Response
The authors describe genetic analysis of several parechoviruses, including recombinant strains. This study adds important information on parechovirus epidemiology in Brazil, as reports from South America are limited. It is a good paper and it should definitely be published after minor revisions. However, the writers should pay attention to details, which are described below.
Specific comments:
Check carefully the nomenclature of viruses and the use of italics in virus name? Species and genus name in italics, virus name not in italics. There are good instructions on the journal pages.Resp; We changed the form of genus and species names.
Human parechovirus species is now called parechovirus-A (PeV-A) and Ljungan virus is PeV-B. Also the proof that Ljungan virus can infect humans remains an open question. HPeV and LjV are still being used in publications, but I suggest using the new names. HPeV-1 => PeV-A1, HPeV-4 => PeV-A4 etc.Resp. All names were updated
The authors should carefully go through the abbreviations: throughout the text human parechoviruses there are several different forms, HPeV or HpeV. These are also in figures (Fig 1). Resp; All fixed
The same is here with genome names: “VP1 gene region”, “Vp1 gene”, “gene VP1”, “Vp1”, all these can be found in the text as well as figure legends. (check also 3D) Resp; Genes and proteins were named accordingly
In general, symbols for genes are italicized (e.g., IGF1), whereas symbols for proteins are not italicized (e.g., IGF1).
Figure 3 is not referred to in the text. Add reference. Resp; Figure 3 now is mentioned in the line 261 of the new manuscript
Abstract:
The first sentence is a bit confusing. Persistence of a virus could mean also that the infection in an individual is persistent or chronic. Here it does not mean this. The first sentence could simply be removed. Resp; The abstract was reduced and this sentence removed.
There is possibly a typo: 24 strains (17 HPeV1; 5 HPeV4; 1 HPeV17, makes only 23). In figure 3 there are 18 HPeV1 types. Resp; We have generated 24 HPeV genomes in this study. We also used HPeV sequences of distinct genotypes to classify our 24 strains
In addition, the number of specific types (18 HPeV1; 5 HPeV4; 1 HPeV17) is only given in figure 1 and the abstract. It should be added to results as well, in the sentences in rows 172-176 this could be specified. Resp we did change this paragraph to “ It also demonstrates that eighteen of our strains (i.e., #09, 21, 43, 47, 48, 50, 53, 54, 65, 66, 82, 88, 110, 158, 170, 175 , 183 and 237) are located within the clade of HPeV genotype 1, another strain (#102) is related to the HPeV reference of genotype 17 and five (17, 23, 74, 80 and 84) fall within the clade formed by genotype 4 strains.”
Introduction:
Poliovirus is not a genera nor a species. Row 50: could be “Enteroviruses, especially (or including) polioviruses, Hepatoviruses …”Resp; This was corrected in the new version of the manuscript.
Materials and methods:
Genebank accession numbers should be included before publication.Resp all 24 strains sequences in this work were submitted to genbank and Ids are pending. Before publication these numbers will be available and included in the manuscript
Results:
Row 184: “strains that belong to the same VP1 genotype”. Should this be HPeV1 genotype? Resp; the sentence was changed to: The tree inferred using the gene 3D has 47% of star-trees and low branch support and there is no clustering pattern of strains.:
Row 229-231 the last sentence of the chapter has the word “some” twice. Resp. this was corrected
Discussion:
Rows 279-283: First two sentences should rather be included in the background, not here.Resp; Sentences removed
What could be mentioned here is that in this study 10% of these diarrhea samples were positive, which is quite high when compare to many other studies. Also, most HPeV positive stool samples are found from slightly younger children, below the age of 2 years. In younger children the detection rate would probably have been somewhat higher. This could be mentioned in discussion even though this is not the main thing in this study. Resp; We did include this percentage of Parechovirus infection in the discussion section. Patients of this study were 0.5 to 2.5 years old when they were affected by gastroenteritis. This information was changed in the new version of the manuscript.
Supplementary figures:
there are two “Figure S2. 3D tree of HPeV”, but the actual figures look different from each other. The latter Should probably be S3? Resp This was corrected
First Figure S2, the color codes in the figure are different from what is said in the legend text. Resp; colors of Brazilian samples were corrected to be match the legend of this figure